# Supermassive Black Holes and Dark Halo from the Bose-Condensed Dark Matter †

**Masahiro Morikawa * and Sakura Takahashi**

Department of Physics Ochanomizu University 2-1-1 Otsuka, Bunkyo, Tokyo 112-8610, Japan; sakura.takahashi@cosmos.phys.ocha.ac.jp

* Correspondence: hiro@phys.ocha.ac.jp

† Presented at the 7th International Conference on New Frontiers in Physics (ICNFP 2018), Crete, Greece, 4–12 July 2018.

**Abstract:** Most of the galaxies harbor supermassive Black Holes (SMBH) in their center. Some of them are observed in very high redshifts. We explore the possibility that SMBH form from the coherent waves of Bose-Einstein condensate (BEC) which are supposed to form the dark matter. We first study the isotropic and anisotropic collapses of BEC. We find the BEC wave can easily collapse to form SMBH but the realistic amount of angular momentum completely prevents the collapse. We further explore the Axion case with attractive interaction and find the moderate mass ratio between the SMBH and the dark halo around it. We also obtain the mass distribution function of BH within a single galaxy.

**Keywords:** SMBH formation; BEC DM/DE; GP equation; Gaussian approximation

## 1. Introduction

Supermassive black holes (SMBH) of mass $10^{6-10}M_\odot$ are observed in the most of the galaxies [1]. All of them are located in the galactic center and not a small amount of SMBHs are already formed very early in the cosmic history of redshift $z \approx 6 - 7.5$ [2]. Moreover, masses of the SMBH have strong correlation with the velocity dispersion $\sigma$ of the galactic bulge $M_{BH} \propto \sigma^4$ [1]. All of these properties strongly indicate that the SMBH plays the central role in the galaxy history and even defines the center of the galaxy [3]. Then the principal question would be the origin of these SMBH. Individual questions related to the SMBH formation will be,

1. (universality in the Universe) Why most of the galaxies harbor SMBH of huge size $10^{6-10}M_\odot$?
2. (location in the galaxy) Why all the SMBH is located at the center of the galaxy?
3. (causal relation with galaxy) Why SMBH form so early at least as $z \approx 6 - 7.5$?
4. (correlation with galaxy) Why SMBH is firmly correlated with the galaxy bulge $M_{BH.} \propto \sigma^4$?
5. (correlation with dark halo (DH)) Why SMBH mass is correlated with the galaxy dark halo mass $M_{DH}$ as $M_{BH.} \approx 10^{-4}M_{DH}$?

There have been many literature in the past trying to answer these questions. Most of them have been generally summarized in the diagram in [4]. Basic mechanism in these is the gas collapse to form the primordial black holes (BH), followed by the coalescence of them or the accretion of the gas on them. Relatively heavy BH is expected from the early population stars of mass $10^{1-2}M_\odot$ [5]. The gas heating tends to prevent the effective accretion. If a huge gas clump directly collapses to form a black hole of size $10^{5-6}M_\odot$ [6], then the Eddington accretion rate can explain the early formation of SMBH although some tuning of parameters is needed.

All of these theories assume that the SMBHs are formed by baryons or fermions. Contrary to them, we now would like to explore another possibility that the SMBHs are formed by the dark matter

(DM) or bosons, in particular, by the quantum condensed boson fields such as the Bose-Einstein condensation (BEC). The quantum condensation of bosons behaves very differently from fermions, or from the fermion condensations, and forms the macroscopic coherent wave. If the DM is formed from light bosons, then it can easily form the quantum condensates, which may coherently collapse to form SMBH in the early stage [7]. Then the stars and their clusters are formed around these SMBH afterward.

Use of these bosonic fields to derive any stable structure in the Universe has a long history of research in the past [8]. We now use the similar boson field model, not for deriving stable structures, but for very unstable case forming black holes.

All the above individual questions naturally suggest that the SMBH defines the center of the galaxy first. SMBH can be formed before stars and galaxies even at $z \approx 10 - 20$, and the SMBH may trigger the subsequent star and galaxy formation at later time $z \leq 10$. In other words, the co-evolution might be very rapid in the early stage and the galaxy merger would not be the dominant mechanism at the SMBH formation stage [9]. We now focus on the first stage of this scenario, i.e., the early formation of SMBH. The subsequent star and galaxy evolution will be discussed in separate articles, extending [3], and including observational predictions.

Thus, we explore in this paper the possibility of the collapse of the coherent boson field which may be the main component of the dark matter (DM) [10]. Our problem is now extended to the question of how SMBH and dark halo (DH), being two different forms of DM, are separated from each other?

In the next Section 2, we clarify how the BEC DM is possible and how the condensation evolves in the Universe. In Section 3, we explore the collapsing dynamics in various conditions and show the BEC DM actually collapses to form SMBH/DH. In Section 4, we consider the Axion model for BEC-DM and try to derive the time and mass scales of the SMBH. In the last Section 5, we conclude our study and describe the subsequent scenario for the galaxy formation triggered by the SMBH.

## 2. How Do SMBHs Form?

We now consider how the cosmic Bose-Einstein condensation (BEC) is possible as dark matter (DM) and dark energy(DE) [10–13]. The critical temperature, below which the BEC takes place, is given by

$$kT_{cr} = \frac{2\pi\hbar^2 n^{2/3}}{\zeta(3/2)^{2/3}m}, \tag{1}$$

where $n$ is the number density of the boson particle of mass $m$ and $\zeta(3/2) \approx 2.6$ is the zeta function. On the other hand, the cosmic DM density evolution is given by

$$n = n_0 \left( \frac{m}{2\pi\hbar^2} \frac{T}{T_0} \right)^{3/2}, \tag{2}$$

where $T$ is the temperature of the DM and the suffix 0 denotes the present time. This expression for $n$ is obtained by the conservation of the entropy $s$ per number density,

$$\frac{s}{n} = \ln \left( \frac{e^{5/2}}{\left(2\pi\hbar^2\right)^{3/2}} \left(\frac{m}{T}\right)^{3/2} \frac{T^3}{n} \right). \tag{3}$$

It is apparent that Equations (1) and (2) have the same proportionality $T \propto n^{2/3}$. Therefore, once the Universe enters into the phase of BEC, it stays in BEC in the later evolution provided the process is adiabatic. The Universe would be mostly adiabatic but locally violated, for example, in the violent process such as the formation of SMBH by BEC. Therefore, for example if the boson temperature was equal to the radiation temperature before the redshift $z = 3000$, $T_{cr} = 0.0027K$, $\rho_0 =$

$9.44 \times 10^{-30}$ g/cm$^3$, then we would expect that the whole DM/DE is in BEC phase if $m \leq 10$ eV. If the boson were the Axion field, then the temperature would be ultra-cold and the BEC is inevitable.

BEC is described by the wave $\psi(t, \mathbf{x})$ which evolves by the nonlinear Schroedinger equation i.e., Gross-Pitaevskii equation (GP) [14,15],

$$i\hbar \frac{\partial \psi(t, \mathbf{x})}{\partial t} = \left( -\frac{\hbar^2}{2m}\Delta + m\phi + g|\psi|^2 \right) \psi, \tag{4}$$

where $g = 4\pi\hbar^2 a_s/m$, $m$ is the boson mass, and $a_s$ is the scattering length, as well as the Poisson equation (PE)

$$\Delta\phi = 4\pi G m |\psi|^2, \tag{5}$$

where $\phi$ is the gravitational potential.

Both the above equations are the Newtonian approximation of the full general relativistic formulation. However, as the first approximation, this approach would be effective to identify the formation of BH by the criterion that the amount of mass $M$ is compressed into the corresponding Schwarzschild radius $r_s = 2GM/c^2$.

## 3. SMBH from BEC

The GP equation can be solved by the standard numerical methods [16,17]. However the simultaneous calculation of the parabolic GP and the elliptical PE equation, is generally difficult to solve in particular for the fast collapsing dynamics in which the solution may not converge to any equilibrium finite form. Here we use semi-analytic calculations for a general argument for the BH formation. Therefore we will further make bold approximations.

### 3.1. Isotropic Collapse

The Lagrangian that yields the GP and PE is given by

$$\begin{aligned} L = & \ (i\hbar/2)\left(\psi^\dagger\dot{\psi} - \dot{\psi}^\dagger\psi\right) - \left(\hbar^2/2m\right)\nabla\psi^\dagger\nabla\psi - (g/2)\left(\psi^\dagger\psi\right)^2 \\ & - (1/8\pi G)\nabla\phi\nabla\phi - m\phi\psi^\dagger\psi. \end{aligned} \tag{6}$$

In order to make the semi-analytic calculations possible, we use the Gaussian approximation [18],

$$\psi(t, x) = Ne^{-r^2/(2\sigma(t))^2 + ir^2\alpha(t)}, \quad \phi(t, x) = -\mu(t)e^{-r^2/(2\tau(t))^2}, \tag{7}$$

where $N$ is the number density of the boson particles. We roughly estimate the BH formation when the portion of the DM enters inside the corresponding Schwarzschild radius.

Integrating this $L$ over the entire three-dimensional space, we have the effective Lagrangian for the relevant variables $\sigma(t), \alpha(t), \mu(\tau)$, and $\tau(t)$,

$$\begin{aligned} L_{\text{eff}} = & \ 1/16(-(2\sqrt{2}gN^2)/(\pi^{3/2}\sigma(t)^3) - (12N\hbar^2)/(m\sigma(t)^2) - (48N\hbar^2\alpha(t)^2\sigma(t)^2)/m \\ & + (32\sqrt{2}N\mu(t))/(\sigma(t)^2(2/\sigma(t)^2 + 1/\tau(t)^2)^{3/2})) \\ & - (3\sqrt{p}\mu(t)^2\tau(t))/G - 24N\hbar\sigma(t)^2\alpha'(t)). \end{aligned} \tag{8}$$

Then we can derive the ordinary differential equation,

$$-\frac{\sqrt{2}gN}{\pi^{3/2}} + \frac{100}{81}\sqrt{\frac{10}{\pi}}Gm^2N\sigma(t)^2 - 6m\sigma(t)^4\sigma''(t) - \frac{6\hbar^2\sigma(t)}{m} = 0, \tag{9}$$

where the phase $\alpha(t)$ and the other variables are dependent variables. It turns out, for the free case $g = 0$, that the effective potential $V_{\text{eff}}(\sigma)$, derived from the Lagrangian Equation (8),

$$V_{\text{eff}}(\sigma) = \frac{gN}{\sigma(t)^3} - \frac{Gm^2N}{\sigma(t)} + \frac{\hbar^2}{m\sigma(t)^2} \tag{10}$$

has a minimum $\sigma_{min}$ beyond which the variable $\sigma(t)$ cannot reduce. The condition that the system of $\sigma(t)$ stays $\sigma_{min}$ forms a BH is $M > M_{kaup}$, where $M$ is the total mass inside the radius of this minimum $\sigma_{min}$ and $M_{kaup} = 0.633\hbar c/(Gm)$ is the Kaup mass. This mass is the critical mass at which the gravity and the quantum pressure balance with each other. If this condition holds, most of the mass turns into a black hole in the Gaussian approximation. A typical numerical solution is given in Figure 1 Left.

*3.2. Anisotropic Collapse*

It is easy to extend the above method to the anisotropic BEC collapse, simply introducing independent dispersion for each spatial directions $\sigma_i(t)$, $i = 1, 2, 3$, and the BEC field is expresses as

$$\psi(t, x) = \exp\left[ix_1^2\alpha_1(t) - \frac{x_1^2}{2\sigma_1(t)^2} + ix_2^2\alpha_2(t) - \frac{x_2^2}{2\sigma_2(t)^2} + ix_3^2\alpha_3(t) - \frac{x_3^2}{2\sigma_3(t)^2}\right]. \tag{11}$$

However, the effective Lagrangian becomes lengthy,

$$L_{\text{eff}} = -\frac{N}{4m\sigma_1(t)^2} - \frac{N\hbar^2}{4m\sigma_2(t)^2} - \frac{N\hbar^2}{4m\sigma_3(t)^2} - \frac{gN^2}{4\sqrt{2}\pi^{3/2}\sigma_1(t)\sigma_2(t)\sigma_3(t)} + \ldots - \frac{1}{4}mN\sigma_3(t)\sigma_3^{''}(t), \tag{12}$$

and the corresponding effective potential is ugly,

$$V_{\text{eff}} = \frac{gN}{2\sqrt{2}\pi^{3/2}\sigma_1(t)\sigma_2(t)\sigma_3(t)} + \frac{25\sqrt{\frac{10}{\pi}}GN\sqrt[3]{\sigma_2(t)}\sqrt[3]{\sigma_3(t)}}{243\sigma_1(t)^{5/3}} + \ldots + \frac{\hbar^2}{2m\sigma_1(t)^2} + \frac{\hbar^2}{2m\sigma_2(t)^2} + \frac{\hbar^2}{2m\sigma_3(t)^2}, \tag{13}$$

though being still useful for a rough analytic estimates and numerical calculations. A typical numerical solution is given in Figure 1 Right. Even in the anisotropic case, BEC can collapse to form SMBH. Unfortunately, at the first collapse, most of the DM turns into SMBH and almost no DH is left behind if we neglect the dissipation or angular momentum.

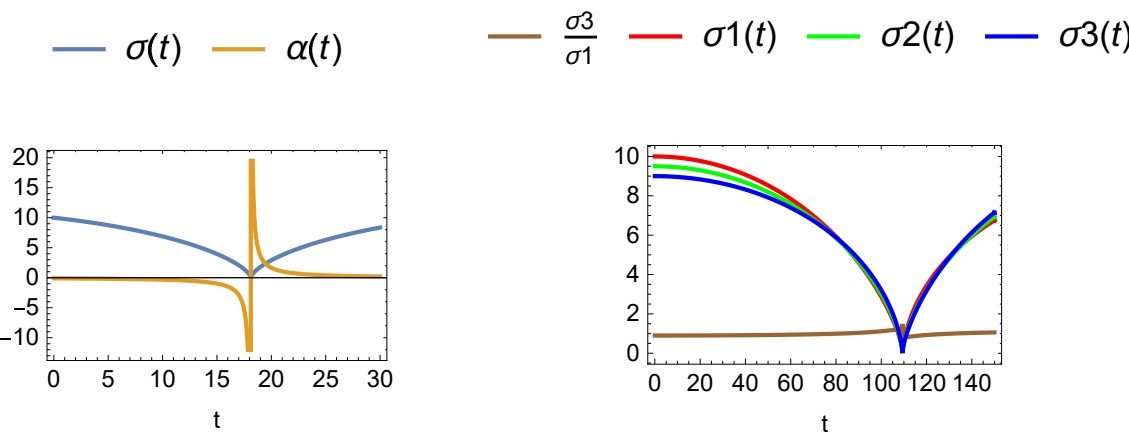

**Figure 1.** Time evolution of the typical collapsing dynamics of BEC. (**Left**) Isotropic collapse. The time evolution of the dispersion $\sigma(t)$ and the phase $\alpha(t)$ of the condensate filed. If the minimum size $\sigma_{min}$ is smaller than the corresponding Schwarzschild radius, a BH is formed, otherwise it simply bounces. (**Right**) Anisotropic collapse. The time evolution of the dispersion $\sigma_i(t)$ and the ratio $\sigma_3(t)/\sigma_1(t)$ of the condensate filed. Anisotropic BEC can easily collapse to form BH as well.

Further, it would be more realistic to consider the dissipative collapse. Usually, the decay of the BEC into the normal gas is expressed by an extra term $\gamma\psi(t,x)$ on the left-hand side of Equation (4), where $\gamma$ is a constant. However, such dissipation cannot be expressed in the proper Lagrangian. We now introduce an explicitly time-dependent factor $e^{\gamma t}$ as

$$L_{\text{diss}} = e^{\gamma t}L \tag{14}$$

where $L$ is the original Lagrangian Equation (6). Integrating this $L_{\text{diss}}$ over the entire space as before, we have the effective Lagrangian which yields the dissipative equation of motion. A typical numerical calculation of this is in Figure 2 Left.

Superposition of the snapshots of BEC field at each maximum expansion is given in (Figure 2 Right). A set of concentric shell structure is conspicuous. The shell may form a temporal potential well, which induce the active star formation there. If this structure is supported, for example, by the rotation of the gas, this may leave some trace such as the concentric shell structure of stars in the later evolution of the galaxy [19].

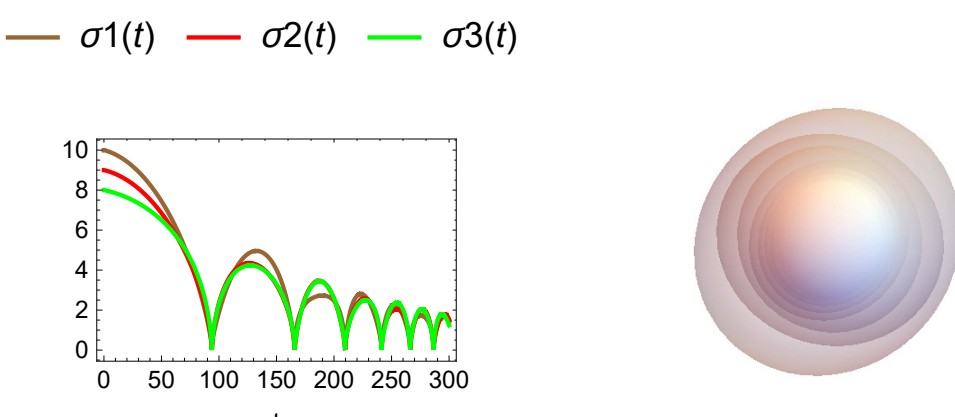

**Figure 2.** Time evolution of a dissipative anisotropic BEC collapse. (**Left**) A damping collapse of BEC. This is a typical solution derived from the effective Lagrangian Equation (14). (**Right**) The superposition of the snapshots of BEC field at each maximum expansion. It forms the concentric almost spherical shells around the formed SMBH.

### 3.3. Collapse with Angular Momentum

The angular momentum of the system generally has a tendency to prevent the BEC collapse into SMBH. However, an appropriate amount of angular momentum may be important to get the coexistence of SMBH and DH, because the whole system would collapse into BH otherwise. In this context, from various observations, the individual angular momentum $A$ of the cosmological objects shows the following scaling [20],

$$A = \kappa\frac{G}{c}M^2, \quad \kappa \approx 10^4. \tag{15}$$

In our case, if we start from the wave function

$$\psi(t,\mathbf{x}) = e^{-\frac{r^2}{2\sigma(t)^2}+ir^2\alpha(t)}Y_l^m(\theta,\phi), \tag{16}$$

the effective potential becomes

$$\frac{gN}{\sigma(t)^3} - \frac{Gm^2N}{\sigma(t)} + \frac{A^2}{m\sigma(t)^2}. \tag{17}$$

Therefore, if $g = 0$, the minimum of this potential yields the maximally allowed angular momentum for the BH formation. It turns out to be $A_{\max} = 2\frac{G}{c}M^2$ which is, unfortunately, well below the observed value Equation (15) and therefore no BH is formed at all.

From this situation, we need to consider much dominant interaction term with $g < 0$, the attractive force which overcomes the prevention of the collapse to SMBH by the angular momentum. There is another possibility that the SMBH is formed in much earlier stage when the over-density region acquires its angular momentum through the tidal torque mechanism [21]. This latter possibility will be studied in a separate article.

## 4. Collapse of Axion Field -Attractive Interaction-

We now consider the attractive interaction ($g < 0$) expecting that this moderately promotes the SMBH formation by reducing the angular momentum effect. A typical boson field with attractive force would be the Axion, which is also a good candidate of DM [22]. The Axion field would be initially uniform and forms BEC because of its low temperature and small mass. We do not know the shape of the developed over-density region, but let us simply assume it as the quasi-isothermal distribution as a toy model,

$$\rho(r) = \rho_0 \left(1 + \left(\frac{r}{r_0}\right)^2\right)^{-1},\tag{18}$$

where the system is supposed to extend up to the radius $R$. We further assume that the rigid rotation of the condensed system with a constant angular velocity $\omega$. Then the angular momentum $J$ of the region inside some radius $r$ is simply given by integrating this rotation with the weight Equation (18) to this radius. We now consider the effective potential Equation (17) for this system. Generally, this potential has a maximum and a minimum,

$$\frac{J^2 \pm \sqrt{J^4 - 48\pi a_s GM^6 m^{-3}\hbar^2}}{2GM^3},\tag{19}$$

where $M$ is the mass inside the radius $r$ (Figure 3 Left).

If these maximum and the minimum coincide with each other, then the barrier of the angular momentum disappears and the corresponding region of BEC can collapse into BH. This radius is estimated as

$$r_{hb} = \frac{2\sqrt{3\pi}a_s^{1/2}\hbar}{G^{1/2}m^{3/2}},\tag{20}$$

which is fully given by microscopic constants and independent from the DM mass and angular momentum $M, J$. Axion attractive force, despite being very weak, just cancels the effective potential barrier formed by the angular momentum at this scale (Figure 3 Right). The part of the system inside of this radius $r_{hb}$ collapses to form SMBH and the rest of the system would form DH surrounding the SMBH.

A typical scales for the Axion and the galaxy

$$m = 10^{-5}\text{eV}, a_s = 10^{-29}\text{meter}, r_0 = 1\text{kpc}, R = 10\text{kpc}\tag{21}$$

yield the values

$$r_{hb} = 108pc, M_{SMBH}/M_{DH} = 4.2 \times 10^{-5}, t_{BH} = 6.3 \times 10^4 year\tag{22}$$

where $M_{SMBH}/M_{DH}$ is the mass ratio of the formed SMBH and the surrounding DH, and $t_{BH}$ is the time scale of the SMBH formation.

Although a SMBH is formed within a very short time scale in this scenario, too many such SMBH would be formed near the center of the galaxy. Because one BH is formed in the volume defined by the

above scale $r_{hb}$ everywhere in the galaxy. Then these dense SMBH soon collide with each other to form larger SMBH. This coalescence process takes place particularly in the central core regions of size $r_0$. Then about $(r_0/r_{hb})^3 \approx 10^3$ SMBH would coalesce with each other to form much larger SMBH at the center of the galaxy.

We can estimate the time scale of this whole process from the evaporation time scale using the standard dynamical friction theory [23]. Assuming the Boltzmann distribution for an *N*-body self-gravitating system, and that the portion of the positive energy particles in the whole distribution, $\gamma \approx 0.0074$, can escape from the cluster to infinity, the time scale becomes

$$\tau_d = \frac{2}{9\gamma}\tau_r \approx 30.1\tau_r, \tag{23}$$

where the dynamical relaxation time scale is

$$\tau_r = \frac{N}{12\sqrt{2}\ln N}\tau_c, \tag{24}$$

and the free-fall time scale is

$$\tau_c = \frac{R}{v} = \sqrt{\frac{R^3}{GM}}. \tag{25}$$

Then Equation (23) yields about $10^8$ years for our case $N \approx 10^3$, well within the observational constraints.

Furthermore, in this Axion case, many smaller black holes, of mass range $10^{2-5}M_\odot$, are formed as well in the outskirts of the galaxy. The mass function of them has almost the power law distribution and one dominant contribution from the SMBH (Figure 4). This is the mass distribution function of BH within a single galaxy and should not be confused with the ordinary global mass function of BH in the Universe usually expressed by the Press-Schechter function. A galaxy turns out to be filled with plenty of black holes of various masses in this case.

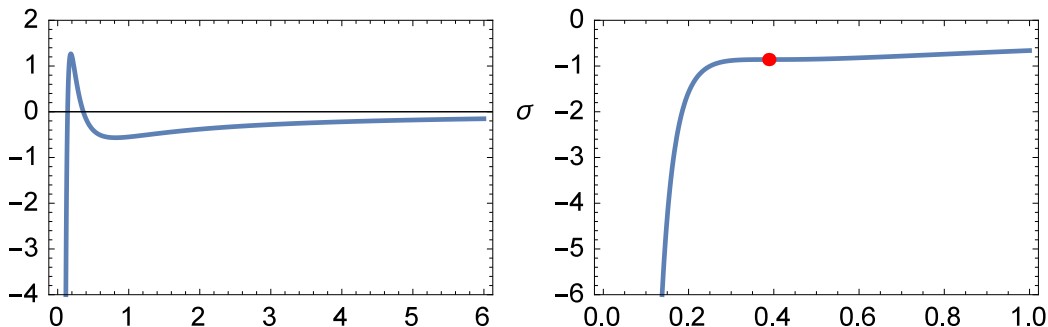

**Figure 3.** The effective potentials for the Axion DM. (**Left**) There is a barrier formed by the angular momentum and a bottom formed by the gravity on its right. (**Right**) The barrier top and the bottom merge at a special scale (marked by a red point) and the BEC portion inside this scale generates SMBH.

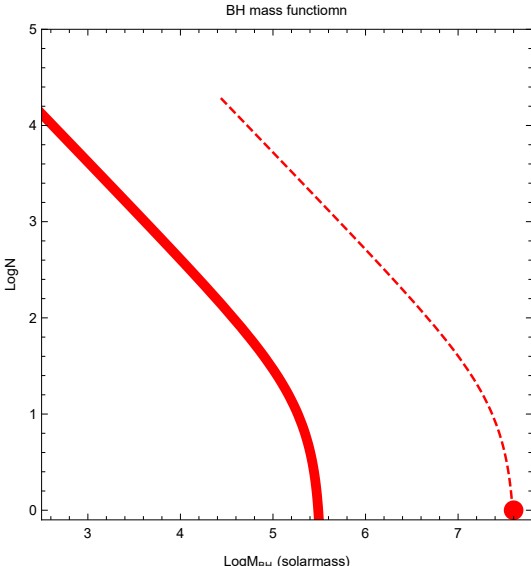

**Figure 4.** The mass distribution functions of BH within a single galaxy. The broken line shows the case for $a_s = 10^{-29}$ m, while the solid line and the separate point show for $a_s = 10^{-30.4}$ m. The latter case represents a more realistic estimate for the SMBH formation considering the coalescence at the central region by the dynamical friction within the time scale $t_d = 6.8 \times 10^7$ years.

## 5. Conclusions and Discussions

We have considered the scenario that the Bose-Einstein condensate (BEC) dark matter (DM) collapses to form supermassive black holes (SMBH) in the galactic center. This coherent wave easily collapses to form SMBH. Actually, by using a simple Gaussian non-relativistic approximation, we have obtain the collapsing dynamics of the isotropic as well as anisotropic distributions of BEC. Thus SMBH is formed but most of the BEC collapses leaving no dark halo (DH) in the surrounding regions in these simple cases. On the other hand, if we introduce the realistic amount of angular momentum, BEC does not collapse at all to form BH and simply the whole DH just remains. Then, we have introduced a small amount of attractive interaction of the condensed bosons, such as Axion field. We have found that some portion near the center of the BEC system collapses to form thousands of BHs which eventually coalesce with each other by the dynamical relaxation and form a single SMBH within the remaining huge amount of DH. A typical mass ratio of them turns out to be $M_{SMBH}/M_{DH} \approx 10^{-4}$. This may yield a commonly observed SMBH-DH correlation. We have obtained a tentative mass distribution function of BH within a single galaxy, which should be compared with future observations.

In order to complete this scenario of SMBH formation from BEC, the following work is needed. Firstly, if we do not rely upon the Axion model, we have to go back to the early Universe when the galaxy/halo first obtained their angular momentum by the tidal torque mechanism. Secondly, we have to study the repeated collapse and bounce of BEC with diminishing amplitude, in relation to the general concentric shell structure observed in many galaxies. Finally, we have to extend our calculation based on the general relativity in order to clarify the precise process and condition of the BH formation.

**Author Contributions:** Conceptualization, M.M.; methodology, M.M.; software, S.T.; validation, M.M., and S.T.; formal analysis, M.M.; investigation, M.M.; resources, S.T.; data curation, S.T.; writing–original draft preparation, M.M.; writing–review and editing, M.M.; visualization, S.T.; supervision, M.M.; project administration, M.M.; funding acquisition, M.M.

**Funding:** This work was supported by JSPS KAKENHI Grant Number KK18K18765.

**Conflicts of Interest:** The authors declare no conflict of interest.

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
