# Peer review of "Supermassive Black Holes and Dark Halo from the Bose-Condensed Dark Matter†"

_proceedings, 2019_

Round 1

Reviewer 1 Report

The revised version takes 2 of my three previous comments into account. The response to my previous first comment is not clear, however. The paper may be published, even if its overall ranking is slightly below average.

Author Response

We’ve checked all the references implied in the above. We further recognized the fact that there are many such references in the past about the boson stars beyond the above-suggested references. It is impossible to cite all of them. Therefore we decided to cite a recent review of the boson stars in which all the relevant references are cited.

L. Liebling and C. Palenzuela, Living Rev Relativ. \textbf{20} 5, 2017.and many references listed.

Reviewer 2 Report

Review of the revised version of "Supermassive black holes and dark halo from the Bose-condensed dark matter" by Morikawa and Takahashi

Basically the authors did not pay attention to what I wrote the last time.

To give more references as regards Bose stars: Ruffini et al. 1969 Phys Rev 187, 1767; Ingrosso et al. 1990 Nuovo Cimento B 105, 977; Kolb et al. 1993 PRL 71, 3051; Kolb et al. 1994 PRD 49, 5040.  The treatment always requires Oppenheimer & Volkoff 1939 Phys Rev 55, 374.  Once you use a generalized treatment of the equation of state, Bose stars and Fermi stars both require Oppenheimer & Volkoff.  Bose stars are often unstable as has been known for at least three decades, and easily become black holes.

The idea that BHs derive from some other form of matter, like dark matter, is really old.  One of the more recent variants of this is due to N. Sanchez, after decades of work by her.

I do not want to just repeat my earlier comments, as they all still apply - the authors just added a few references to some ideas. To give the conclusion.

Author Response

Our previous answer is just what we can provide at best at least at the present stage

We simply introduced the quartic interaction term of the Axion from the expansion of the original cosine potential. The cubic interaction motivated from the other source may be interesting for SMBH formation. However, at present, we do not know how we treat the instability of the cubic interaction which has no lower bound in the potential energy.

Round 2

Reviewer 2 Report

Review of the twice revised version of "Supermassive black holes and dark halo from the Bose-condensed dark matter" by Morikawa and Takahashi

Basically the authors did not pay attention to anything what I wrote the last two times.

Without using Oppenheimer & Volkoff all this is naive. They do not even mention this pivotal paper. One needs to use the equations of Oppenheimer & Volkoff (1939), really well established. Once given this frame work one can argue about the equation of state, and what goes into it, doing all BEC arguments which one wants; but most of this has been done already, in a different language. This paper is just not a useful enterprise, as it basically goes back to before 1939, some would argue before 1932 (Landau, Chandrasekhar).

This manuscript is a resubmission of an earlier submission. The following is a list of the peer review reports and author responses from that submission.

Round 1

Reviewer 1 Report

Review of "Supermassive black holes and dark halo from the Bose-condensed dark matter" by Morikawa and Takahashi It is a really old idea to use quantum fluid properties of very dense particles, whatever their nature. Modern work related to such an approach has been done, e.g., by Munyaneza & Viollier, by Sanchez & de Vega, and others, with some ideas based on Oppenheimer & Volkoff 1939 early on. The arguments surrounding eqs 3 - 5 are also quite old, and actually wrong in the sense, that it has been shown already in the 1960s by Spitzer and his group, that you can make a SMBH of order 10^6 solar masses relatively quickly, and then do the rest by accretion. The standard cosmological simulations manage to obtain reasonably early SMBH masses of the required numbers, once you consider some subtleties in cooling. So these authors here overstate their case, and their arguments are not new. Some of the arguments about merging BHs have been made Spitzer, Rees, and many others, many of these papers were done in the 1960s and 1970s. The arguments right after eq 25 seem to invalidate the idea proposed. The mass function of SMBHs has been published a long time ago. It does not look like Fig 4. It can readily be described as a Press-Schechter function, i.e a power-law with an exponential cutoff. The kink is just above 10^8 solar masses. The low cut is around a few million solar masses. There are truncated galaxies, for which the black hole mass dark halo relation no longer holds, possibly due to tidal effects of an even more massive neighbor. Radio observations of radio galaxies strongly suggest that SMBHs grow by merging. In some cases one can observe the sequence of merger events (see papers by the group around A. Bogdan). This paper is basically a mathematical exercise with little relation to observation. As a mathematical exercise it is a variation of old ideas. What it does predict, such as the mass function, is discrepant. To be sufficiently interesting to be published, the analogies to older work have to be acknowledged, and the connection to observation needs to be made very much tighter - that part will be hard work. Finally, there better be some prediction, what would follow from this specific version of some really old ideas. But a prediction that is actually true! The modern observations of the SMBH in the center of our galaxy might provide some test. This paper should not be published as is.

Reviewer 2 Report

It has been a longterm hypothesis that a coherent bosonic scalar field was responsible for DM formation. This study deals with such model, with partly analytical results. It may be published after consideration of some minor but important questions:

(1) in a BEC the cubic nonlinearity as well motivated by local 2-body interactions. Does it make sense to include such a term in this context? It rather seems to go back again to a 2 particle scattering model, inconsistent with the self-gravitating potential term! This aspect and the model investigated should be motivated in detail.

(2) Eq. (7) could be solved by standard numerical methods, see e-g Phys. Rev. Lett. 94, 130404 (2005) and Phys. Rev. A 72, 063610 (2005) for a BEC context. Maybe this can be stated somewhere and those/some references be provided.

(3) Pleas add the following standard reference in this field which compares many-body classical simulations with results from the scalar-field model: Nature Physics volume 10pages 496499 (2014).

Reviewer 3 Report

The Authors model the evolution of a supermassive black hole using a coupled system of equations composed from Gross-Pitaevskii equation describing a Bose condensate and a Poisson equation.

I do not find that the proposed model is well justified. Furthermore, the overall level of the presentation is very bad and is not compatible with the high level of the journal.

I do not recommend publication of the present Manuscript.